# Stochastic Approximation of Gaussian Free Energy for Risk-Sensitive Reinforcement Learning

## Abstract

We introduce a stochastic approximation rule for estimating the free energy from i.i.d. samples generated by a Gaussian distribution with unknown mean and variance. The rule is a simple modification of the Rescorla-Wagner rule, where the (sigmoidal) stimulus is taken to be either the event of over- or underestimating a target value. Since the Gaussian free energy is known to be a certainty-equivalent sensitive to the mean and the variance, the learning rule has applications in risk-sensitive decision-making. In particular, we show how to use the rule in combination with the temporal-difference error in order to obtain risk-sensitive, model-free reinforcement learning algorithms.

## 1 Introduction

**Main contribution.** Let $N(x; \mu, \rho) = \sqrt{\frac{\rho}{2\pi}} \exp\{-\frac{\rho}{2}(x-\mu)^2\}$ be the Gaussian pdf with mean $\mu$ and precision $\rho$. Given a sequence $x_1, x_2, \ldots$ of i.i.d. samples drawn from $N(x; \mu, \rho)$ with unknown $\mu$ and $\rho$, consider the problem of estimating the free energy $\mathbf{F}_\beta$ for a given inverse temperature $\beta \in \mathbb{R}$, that is

$$\mathbf{F}_\beta = \frac{1}{\beta} \log \int_\mathbb{R} N(x; \mu, \rho) \exp\{\beta x\} \, dx = \mu + \frac{\beta}{2\rho}. \tag{1}$$

This paper shows that (1) can be estimated using a surprisingly simple stochastic approximation rule. If $v \in \mathbb{R}$ is the current estimate and a new sample $x$ arrives, update $v$ according to

$$v \leftarrow v + 2\alpha \cdot \sigma_\beta(x-v) \cdot (x-v), \tag{2}$$

where $\alpha > 0$ is a learning rate and $\sigma_\beta(z)$ is the scaled logistic sigmoid

$$\sigma_\beta(z) = \frac{1}{1 + \exp\{-\beta z\}}. \tag{3}$$

The unique and stable fixed point of the learning rule (2) is equal to the desired free energy value $v^* = \mu + \frac{\beta}{2\rho}$.

**Motivation.** Risk-sensitivity, the susceptibility to the higher-order moments of the return, is necessary for the real-world deployment of AI agents. Wrong assumptions, lack of data, misspecification, limited computation, and adversarial attacks are just a handful of the countless sources of unforeseen perturbations that could be present at deployment time. Such perturbations can easily destabilize risk-neutral policies, because they only focus on maximizing expected return while entirely neglecting the variance. This poses serious safety concerns (Russell et al., 2015; Amodei et al., 2016; Leike et al., 2017).

Risk-sensitive control has a long history in control theory (Coraluppi, 1997) and is an active area of research within reinforcement learning (RL). There are multiple different approaches to risk-sensitivity in RL: for instance in *Minimax RL*, inspired by classical robust control theory, one derives a conservative worst-case policy over MDP parameter intervals (Nilim and El Ghaoui, 2005; Tamar et al., 2014); and the more recent *CVaR approach* relies on using the conditional-value-at-risk as a robust performance measure (Galichet et al., 2013; Cassel et al., 2018). We refer the reader to García and Fernández (2015) for a comprehensive overview. Here we focus on one of the earliest and most popular approaches (see references), consisting of the use of exponentially-transformed values, or equivalently, the free energy as the risk-sensitive certainty-equivalent (Bellman, 1957; Howard and Matheson, 1972).

The certainty-equivalent of a stochastic value $X \in \mathbb{R}$ is defined as the representative deterministic value $v \in \mathbb{R}$ that a decision-maker uses as a summary of $X$ for valuation purposes. To illustrate, consider a first-order Markov chain over discrete states $\mathcal{S}$ with transition kernel $P(s'|s)$, state-emitted rewards $R(s) \in \mathbb{R}$, and discount factor $\gamma \in [0, 1)$. Typically RL methods use the expectation as the certainty-equivalent of stochastic transitions (Bertsekas and Tsitsiklis, 1995; Sutton and Barto, 2018). Therefore they compute the value $V(s)$ of the current state $s \in \mathcal{S}$ by (recursively) aggregating the future values through their expectation, e.g.

$$V(s) = \int P(s'|s)\{R(s') + \gamma V(s')\}\, ds'. \tag{4}$$

Instead, Howard and Matheson (1972) proposed using the free energy as the certainty-equivalent, that is,

$$V(s) = F_\beta(s) = \frac{1}{\beta} \log \int P(s'|s) \exp\{\beta[R(s') + \gamma V(s')]\}\, ds', \tag{5}$$

where $\beta \in \mathbb{R}$ is the inverse temperature parameter which determines whether the aggregation is risk-averse ($\beta < 0$), risk-seeking ($\beta > 0$), or even risk-neutral as a special case ($\beta = 0$). Indeed, if the future values are bounded, then $F_\beta(s)$ is sigmoidal in shape as a function of $\beta$, with three special values given by

$$\lim_\beta F_\beta(s) = \begin{cases} \min_{s'}\{R(s') + \gamma V(s')\} & \text{if } \beta \to -\infty; \\ \mathbf{E}[R(S') + \gamma V(S')|S = s] & \text{if } \beta \to 0; \\ \max_{s'}\{R(s') + \gamma V(s')\} & \text{if } \beta \to +\infty. \end{cases} \tag{6}$$

These limit values highlight the sensitivity to the higher-order moments of the return. Because of this property, the free energy has been used as the certainty-equivalent for assessing the value of both actions and observations under limited control and model uncertainty respectively, each effect having their own inverse temperature. The work by Grau-Moya et al. (2016) is a demonstration of how to incorporate multiple types of effects in MDPs.

The present work addresses a longstanding problem pointed out by Mihatsch and Neuneier (2002). An advantage of using expectations is that certainty-equivalents such as (4) are easily estimated using stochastic approximation schemes. For instance, consider the classical Robbins-Monro update (Robbins and Monro, 1951)

$$v \leftarrow v + \alpha \cdot (x - v) \tag{7}$$

where $x \sim P(x)$ is a stochastic target value, $\alpha$ is a learning rate, and $v$ is the estimate of $\mathbf{E}[X]$. Substituting $x = R(s') + \gamma V(s')$ and $v = V(s)$ leads to the popular TD(0) update (Sutton and Barto, 1990):

$$V(s) \leftarrow V(s) + \alpha(R(s') + \gamma V(s') - V(s)). \tag{8}$$

However, there is no model-free counterpart for estimating free energies (5) under general unknown distributions. The difficulty lies in that model-free updates rely on single (Monte-Carlo) unbiased samples, but these are not available in the case of the free energy due to the log-term on the r.h.s. of (5). This shortcoming led Mihatsch and Neuneier (2002) to propose the alternative risk-sensitive learning rule

$$v \leftarrow v + \alpha \cdot u \cdot (x - v), \qquad \text{where } u = \begin{cases} (1 - \kappa) & \text{if } (x - v) \geq 0 \\ (1 + \kappa) & \text{if } (x - v) < 0 \end{cases} \tag{9}$$

and where $\kappa \in [-1; 1]$ is a risk-sensitivity parameter. While the heuristic (9) does produce risk-sensitive policies, these have no formal correspondence to free energies.

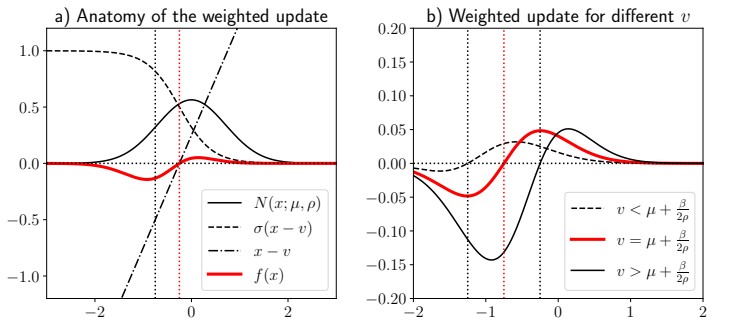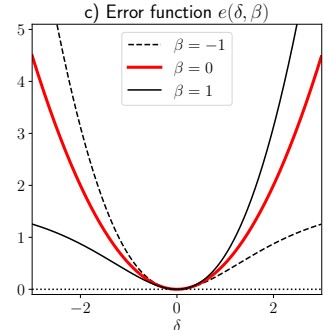

Figure 1: Update rule and its error function. **a)** shows the update $f(x)$ to the estimate $v$ caused by the arrival of a sample $x$, weighted by its probability density. The expected update is determined by comparing the integrals of the positive and negative lobes. **b)** Illustration of weighted update functions $f(x)$ for different values of the current estimate $v$. The positive lobes are either larger, equal, or smaller than the negative lobes for a $v$ that is either smaller, equal, or larger than the free energy respectively. **c)** Error function implied by the update rule. For a risk-neutral ($\beta = 0$) estimator the error function is equal to the quadratic error $e(\delta, 0) = \frac{1}{2}\delta^2$. For a risk-averse estimator ($\beta < 0$), the error function is lopsided, penalizing under-estimates stronger than over-estimates. Furthermore, $e(\delta, \beta)$ is an even function in $\beta$.

As anticipated, our work contributes a simple model-free rule for estimating the free energy in the special case of Gaussian distributions. Starting from the Rescorla-Wagner rule

$$v \leftarrow v + \alpha \cdot u \cdot (x - v), \tag{10}$$

where $u \in \{0, 1\}$ is an indicator function marking the presence of a stimulus (Rescorla, 1972), we substitute $u$ by twice the soft-indicator function $\sigma_\beta(x - v)$ of (3), which activates whenever $v$ either over- or underestimates the target value $x$, depending on the sign of the risk-sensitivity parameter $\beta$. Using the substitutions appropriate for RL, we obtain the risk-sensitive TD(0)-rule

$$V(s) \leftarrow V(s) + 2\alpha \cdot \sigma_\beta(\delta) \cdot \delta, \tag{11}$$

where $\delta = R(s') + \gamma V(s') - V(s)$ is the standard temporal-difference error. The learning rule is trivial to implement, works as stated for tabular RL, and is easily adapted to the objective functions of deep RL methods (Mnih et al., 2015). Finally, the learning rule is also consistent with findings in computational neuroscience (Niv et al., 2012), e.g. predicting asymmetric updates that are stronger for negative prediction errors in the risk-averse case (Gershman, 2015).

## 2 Analysis of the Learning Rule

Our central result is the following lemma, which implies that the unique and stable fixed point of the expected learning dynamics of (2) is given by the desired free energy.

**Lemma 1.** *If $x_1, x_2, \ldots$ are i.i.d. samples from $P(X) = N(x; \mu, \rho)$, then the expected update $J(v)$ of the learning rule* (2) *is twice differentiable and such that*

$$J(v) = 2\mathbf{E}\big[\sigma_\beta(X - v) \cdot (X - v)\big] \begin{cases} < 0, & \textit{if } v > \mathbf{F}_\beta; \\ = 0, & \textit{if } v = \mathbf{F}_\beta; \\ > 0, & \textit{if } v < \mathbf{F}_\beta. \end{cases}$$

*Proof.* The expected update of $v$ is

$$J(v) := 2 \int N(x; \mu, \rho)\sigma(x - v)(x - v)\, dx, \tag{12}$$

where we have dropped the subscript $\beta$ from $\sigma_\beta$ for simplicity. Using the Leibnitz integral rule it is easily seen that this function is twice differentiable w.r.t. $v$, because the integrand is a product of twice differentiable functions.

The resulting update direction will be positive if the integral over the positive contributions outweight the negative contributions and vice versa. The integrand of (12) has a symmetry property: splitting the domain of integration $\mathbb{R}$ into $(-\infty; v]$ and $(v; +\infty)$, using the change of variable $\delta = x - v$, and recombining the two integrals into one gives

$$J(v) := 2 \int_0^\infty \Big\{ N(v + \delta; \mu, \rho)\sigma(\delta) - N(v - \delta; \mu, \rho)\sigma(-\delta) \Big\} \delta \, d\delta. \tag{13}$$

We will show that the integrand of (13) is either negative, zero, or positive, depending on the value of $v$. Define the weighted update $f(x)$ as

$$f(x) = f(v + \delta) := N(v + \delta; \mu, \rho)\sigma(\delta)\delta.$$

This function is illustrated in Figure 1a. We are interested in the ratio

$$\frac{f(v + \delta)}{f(v - \delta)} = \frac{N(v + \delta; \mu, \rho)}{N(v - \delta; \mu, \rho)} \frac{\sigma(\delta)}{\sigma(-\delta)}, \tag{14}$$

which compares the positive against the negative contributions to the integrand in (13). The first fraction of the r.h.s. of (14) is equal to

$$\frac{N(v + \delta; \mu, \delta)}{N(v - \delta; \mu, \rho)} = \exp\Big\{ -\frac{\rho}{2}(v + \delta - \mu)^2 + \frac{\rho}{2}(v - \delta - \mu)^2 \Big\} = \exp\{ -2\rho\delta(v - \mu) \}.$$

Using the symmetry property $\sigma(\delta) = 1 - \sigma(-\delta)$ of the logistic sigmoid function, the second fraction can be shown to be equal to

$$\frac{\sigma(\delta)}{\sigma(-\delta)} = \frac{\sigma(\delta)}{1 - \sigma(\delta)} = \exp\{\beta\delta\}.$$

Substituting the above back into (14) results in

$$\frac{f(v + \delta)}{f(v - \delta)} = \exp\{ -2\rho\delta(v - \mu) + \beta\delta \} \begin{cases} > 1 & \text{for } v < \mu + \frac{\beta}{2\rho}, \\ = 1 & \text{for } v = \mu + \frac{\beta}{2\rho}, \\ < 1 & \text{for } v > \mu + \frac{\beta}{2\rho}, \end{cases}$$

also illustrated in Figure 1b. Therefore, the integrand in (13) is either positive ($v < \mu + \frac{2}{2\rho}$), zero ($v = \mu + \frac{\beta}{2\rho}$), or negative ($v > \mu + \frac{\beta}{2\rho}$), allowing to conclude the claim of the lemma. $\qquad\square$

## 3  Additional Properties

We discuss additional properties in order to strengthen the intuition and to clarify the significance of the learning rule; some practical implementation advice is given at the end.

**Associated free energy functional.**  The Gaussian free energy $\mathbf{F}_\beta$ in (1) is formally related to the valuation of risk-sensitive portfolios used in finance (Markowitz, 1952). It is well-known that the free energy is the extremum of the *free energy functional*, defined as the Kullback-Leibler-regularized expectation of $X$:

$$F_\beta\big[p(x)\big] := \mathbf{E}_p[X] - \frac{1}{\beta}\text{KL}\big(p(x)\big\|N(x; \mu, \rho)\big). \tag{15}$$

This functional is convex in $p$ for $\beta < 0$ and concave for $\beta > 0$. Taking either the minimum (for $\beta < 0$) or maximum (for $\beta > 0$) w.r.t. $p(x)$ yields

$$\mathbf{F}_\beta = \underset{p(x)}{\text{extr}}\, F_\beta\big[p(x)\big] = \Big[\mu + \frac{\beta}{\rho}\Big] - \frac{1}{\beta}\Big[\frac{\beta^2}{2\rho}\Big] = \mu + \frac{\beta}{2\rho} = \mathbf{E}[X] + \frac{\beta}{2}\mathbf{Var}[X], \tag{16}$$

that is, the Gaussian free energy is a linear function of $\beta$, where the intercept and the slope are equal to the expectation and half of the variance of $X$ respectively. The extremizer $p^*(x)$ is the Gaussian

$$p^*(x) = \arg\underset{p(x)}{\text{extr}}\, F_\beta\big[p(x)\big] = N(x; \mu + \tfrac{\beta}{\rho}, \rho). \tag{17}$$

The above gives a precise meaning to the free energy as a certainty-equivalent. The choice of a non-zero inverse temperature $\beta$ reflects a distrust in the reference probability density $N(x; \mu, \rho)$ as a reliable model for $X$. Specifically, the magnitude of $\beta$ quantifies the degree of distrust and the sign of $\beta$ indicates whether it is an under- or overestimation. This distrust results in using the extremizer (17) as a robust substitute for the original reference model for $X$.

**Game-theoretic interpretation.**    In addition to the above, previous work (Ortega and Lee, 2014; Eysenbach and Levine, 2019; Husain et al., 2021) has shown that the free energy functional has an interpretation as a two-player game which characterizes its robustness properties. Following Ortega and Lee (2014), computing the Legendre-Fenchel dual of the KL regularizer yields an equivalent adversarial re-statement of the free energy functional (15), which for $\beta > 0$ is given by

$$\max_{p(x)} \min_{c(x)} \left\{ \int p(x)[x - c(x)] \, dx + \int N(x; \mu, \rho) \exp\{\beta c(x)\} \, dx, \right\}, \tag{18}$$

where the perturbations $c(x) \in \mathbb{R}$ are chosen by an adversary (Note: for the case $\beta < 0$ one obtains a Minimax problem over $p(x)$ and $c(x)$ rather than a Maximin). From this dual interpretation, one sees that the distribution $p(x)$ is chosen as if it were maximizing the expected value of $x' = x - c(x)$, the adversarially perturbed version of $x$. In turn, the adversary attempts to minimize $x'$, but at the cost of an exponential penalty for $c(x)$. More precisely, given the distribution $p(x)$, the adversarial best-response (ignoring constants) is

$$c^*(x) \overset{(a)}{=} \frac{1}{\beta} \log \frac{p(x)}{N(x; \mu, \rho)} \overset{(b)}{=} \frac{1}{2\beta} \left\{ \rho(x - \mu)^2 - \bar{\rho}(x - \bar{\mu})^2 + \log \frac{\bar{\rho}}{\rho} \right\} \overset{(c)}{=} x - \mathbf{F}_\beta, \tag{19}$$

where the equality (a) is true for any choice of $p(x)$; (b) holds if $p(x) = N(x; \bar{\mu}, \bar{\rho})$ for some mean $\bar{\mu}$ and precision $\bar{\rho}$; and where (c) holds if $p(x)$ is the extremizer (17). Here we see that the adversarial perturbations can be arbitrarily bad if $p(x)$ is not chosen cautiously: for instance, for the (Gaussian) Dirac delta

$$p(x) = N(x; \mu, \bar{\rho}) \xrightarrow{\bar{\rho} \to \infty} \delta(x = \mu) \quad \text{we get} \quad c^*(x) = \mathcal{O}\left( \log \frac{\bar{\rho}}{\rho} \right) \xrightarrow{\bar{\rho} \to \infty} +\infty. \tag{20}$$

**Error function.**    Let $\delta = x - v$ be the instantaneous difference between the sample and the estimate. If the update rule (2) corresponds to a stochastic gradient descent step, then what is the error function? That is, if

$$v \leftarrow v - \alpha \cdot \nabla_\delta e(\delta, \beta) = v + 2\alpha \cdot \sigma_\beta(\delta) \cdot \delta,$$

then what is $e(\delta, \beta)$? Integrating the gradient $\nabla_\delta e(\delta, \beta)$ with respect to $\delta$ gives

$$e(\delta, \beta) = 2 \int \sigma(\delta) \delta \, d\delta = \frac{2\delta}{\beta} \log(1 + \exp\{\beta\delta\}) + \frac{2}{\beta^2} \mathrm{li}_2(-\exp\{\beta\delta\}) + \frac{\pi^2}{6\beta^2}, \tag{21}$$

where $\log(1 + \exp(z))$ is the *softplus function* (Dugas et al., 2001) and $\mathrm{li}_2(z)$ is *Spence's function* (or dilogarithm) defined as

$$\mathrm{li}_2(z) = -\int_0^z \frac{\log(1 - z)}{z} \, dz,$$

and where the constant of integration $\frac{\pi^2}{6\beta^2}$ was chosen so that $\lim_{\delta \to 0} e(\delta, \beta) = 0$ for all $\beta \in \mathbb{R}$. This error function is illustrated in Figure 1c for a handful of values of $\beta$. In the limit $\beta \to 0$, the error function becomes:

$$\lim_{\beta \to 0} e(\delta, \beta) = \frac{1}{2}\delta^2,$$

thus establishing a connection between the quadratic error and the proposed learning rule.

**Practical considerations.**    The free energy learning rule (2) can be implemented as stated, for instance either using constant learning rate $\alpha > 0$ or using an adaptive learning rate $\alpha_t > 0$ fulfilling the Robbins-Monro conditions $\sum_t \alpha_t > 0$ and $\sum_t \alpha_t^2 < \infty$.

A problem arises when most of the data falls within the near-zero saturated region of the sigmoid, which can occur due to an unfortunate initialization of the estimate $v$. Since then $\sigma_\beta(x - v) \approx 0$ for most $x$, learning can be very slow. This problem can be mitigated using an affine transformation of the sigmoid that gaurantees a minimal rate $\eta > 0$, such as

$$\tilde{\sigma}_\beta(z) = \eta + (1 - 2\eta)\sigma_\beta(z), \tag{22}$$

which re-scales the sigmoid within the interval $[\eta, 1 - \eta]$. We have found this adjustment to work well for $|\beta| \approx 0$, especially when it is only used during the first few iterations.

If one wishes to use the learning rule in combination with gradient-based optimization (as is typical in a deep learning architecture), we do not recommend using the error function (21) directly. Rather,

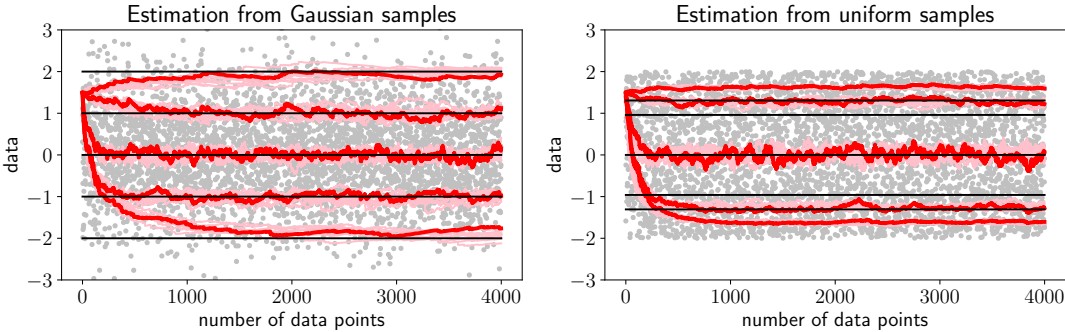

Figure 2: Estimation of the free energy from Gaussian (left panel) and uniform samples (right panel). Each plot shows 10 estimation processes (9 in pink, 1 in red) per choice of the inverse temperature, where $\beta \in \{-4, -2, 0, 2, 4\}$. The true free energies are shown in black. The estimation of the free energy is accurate for Gaussian data but biased for uniform data.

we suggest absorbing the factor $2\tilde{\sigma}_\beta(\delta)$ directly into the learning rate (where as before, $\delta = x - v$). A simple way to achieve this consists in scaling the estimation error $E(\delta)$ by said factor using a stop-gradient, that is,

$$\tilde{E}(\delta) := \text{StopGrad}(2\tilde{\sigma}_\beta(\delta)) \cdot E(\delta), \tag{23}$$

since then the error gradient with respect to the model parameters $\theta$ will be

$$\nabla_\theta \tilde{E}(\delta) = -2\tilde{\sigma}_\beta(\delta) \cdot \frac{\partial E}{\partial \delta} \frac{\partial v}{\partial \theta}. \tag{24}$$

Finally, a large $|\beta|$ chooses a target free energy within a tail of the distribution, leading to slower convergence. If one wishes to approximate a free energy that sits at $n$ standard deviations from the mean, then $\beta$ should be chosen as

$$\beta(n) = 2n\sqrt{\rho}. \tag{25}$$

However, since $\beta(n)$ is not scale invariant and the scale $\rho$ is unknown, a good choice of $\beta$ must be determined empirically.

## 4 Experiments

**Estimation.** Our first experiment is a simple sanity check. We estimated the free energy in an online manner using the learning rule (2) from data generated by two i.i.d. sources: a standard Gaussian, and uniform distribution over the interval $[-2, 2]$. Five different inverse temperatures were used ($\beta \in \{-4, -2, 0, 2, 4\}$). For each condition, we ran ten estimation processes from 4000 random samples using the same starting point ($v = 1.5$). The learning rate was constant and equal to $\alpha = 0.02$.

The results are shown in figure 2. In the Gaussian case, the estimation processes successfully stabilize around the true free energies, with processes having larger $|\beta|$ converging slower, but fluctuating less. In the uniform case, the estimation processes do not settle around the correct free energy values for $\beta \neq 0$; however, the found solutions increase monotonically with $\beta$. These results validate the estimation method only for Gaussian data, as expected.

**Reinforcement learning.** Next we applied the risk-sensitive learning rule to RL in a simple grid-world. The goal was to qualitatively investigate the types of policies that result from different risk-sensitivities. Shown in Figure 3a, the objective of the agent is to navigate to a terminal state containing a reward pill within no more than 25 time steps while avoiding the water. The reward pill delivers one reward point upon collection, whereas standing in the water penalizes the agent with minus one reward point per time step. In addition, there is a very strong wind: with 50% chance in each step, the wind pushes the agent one block in a randomly chosen cardinal direction.

We trained R2D2 (Kapturowski et al., 2018) agents with the risk-sensitive cost function (23) using five uniformly spaced inverse temperatures $\beta$ ranging from $-0.8$ to $0.8$. The architecture of our

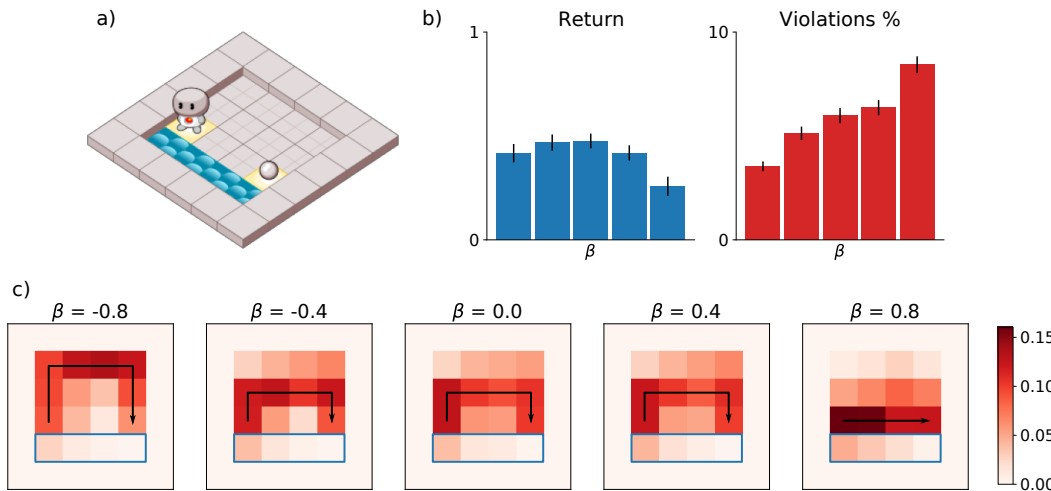

Figure 3: Comparison of risk-sensitive RL agents. **a)** The task consists in picking up a reward located at the terminal state while avoiding stepping into water. A strong wind pushes the agent into a random direction 50% of the time. **b)** Bar plots showing the average return (blue) and the percentage of violations (red) for each policy, ordered from lowest to highest $\beta$. **c)** State visitation frequencies for each policy, plus the optimal (deterministic) policy when there is no wind (black paths).

agents consisted of a first convolutional layer with 3-by-3-kernels and 128 channels, a dense layer with 128 units, and a logit layer for the four possible actions (i.e. walking directions). The discount factor was set to $\gamma = 0.95$. Each agent was trained for 500K iterations with a batch size of 64, using the Adam optimizer with learning rate $10^{-4}$ (Kingma and Ba, 2014). The target network was updated every 400 steps. The inputs to the network were observation tensors of binary features representing the 2D board. Note these agents didn't use any recurrent cells and therefore no backpropagation through time was used. To train all the agents in this experiment we used 154 CPU core hours at 2.4 GHz and 22.5 GPU hours.

To analyze the resulting policies, we computed the episodic returns and the percentage of time the agents spent in the water (i.e. the "violations") from 1000 roll-outs. The results, shown in Figure 3b, reveal that the risk-neutral policy ($\beta = 0$) has the highest average return. However, the percentage of violations increases monotonically with $\beta$. Figure 3c shows the state-visitation probabilities estimated from the same roll-outs. There are essentially three types of policies: risk-averse, taking the longest path away from the water; risk-neutral, taking middle path; and risk-seeking, taking the shortest route right next to the water. These are even more crisply revealed when the wind is de-activated. Interestingly, the risk-averse policy ($\beta = -0.8$) does not always reach the goal, which explains why its return is slightly lower in spite of committing fewer violations.

**Bandits.** In the last experiment we wanted to observe the premiums that risk-sensitive agents are willing to pay when confronted with a choice between a certain and a risky option. To do so, we used a two-arm bandit setup, where one arm ("certain") delivered a fixed reward and the other arm ("risky") a stochastic one—more precisely, drawn from a Gaussian distribution with mean $\mu$ and precision $\rho = 2$. Both the fixed payoff and the mean $\mu$ of the risky arm were drawn from a standard Gaussian distribution at the beginning of an episode, which lasted twenty rounds. To build agents that can trade off exploration versus exploitation, we used memory-based meta-learning (Wang et al., 2016; Santoro et al., 2016), which is known to produce near-optimal bandit players (Ortega et al., 2019; Mikulik et al., 2020).

We meta-trained five R2D2 agents using risk-sensitives $\beta \in \{-1.0, -0.5, 0, 0.5, 1.0\}$ on the two-armed bandit task distribution (also randomizing the certain/risky arm positions) with discount factor $\gamma = 0.95$. The network architecture and training parameters were as in the previous RL experiment, with the difference that the initial convolutional layer was replaced with a dense layer and an LSTM layer having 128 memory cells (Hochreiter and Schmidhuber, 1997). We used backpropagation through time for computing the episode gradients. The input to the network consisted of the action

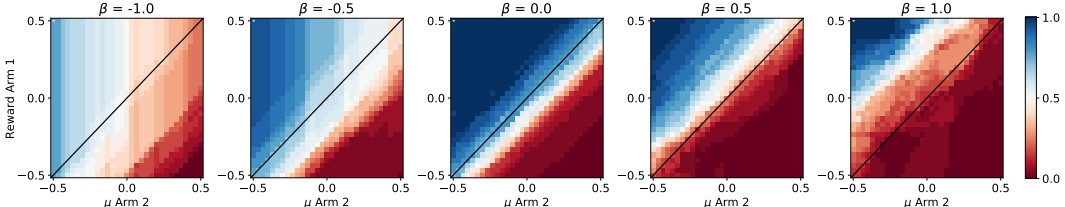

Figure 4: Two-armed bandit policy profiles with different risk-sensitivities $\beta$. The certain arm 1 pays a deterministic reward, while the risky arm 2 pays a stochastic reward drawn from $N(r; \mu, \rho)$ with precision $\rho = 2$. The agents were meta-trained on bandits where the payoffs (i.e. arm 1's payoff and arm 2's mean) were drawn from a standard Gaussian distribution. The plots show the marginal probability of choosing the certain arm (blue) over the risky arm (red) after twenty interactions for every payoff combination. Each point in the uniform grid was estimated from 30 seeds. Note the deviations from the true risk-neutral indifference curve (black diagonal).

taken and reward obtained in the previous step. This setup allows agents to adapt their choices to past interactions throughout an episode. To train all the agents in this experiment we used 88 CPU core hours at 2.4 GHz and 10 GPU hours.

Figure 4 shows the agents' choice profile in the last ($20^{\text{th}}$) time step. A true risk-neutral agent does not distinguish between a certain and risky option that have the same expected payoff (black diagonal). The main finding is that the indifference region (i.e. close to a 50% choice in white color) evolves significantly with increasing $\beta$, implying that the agents with different risk attitudes are indeed willing to pay different risk premia (measured as the vertical distance of the indifference region from the diagonal). We observe two effects. The most salient effect is that the indifference region mostly moves from being beneath (risk-averse) to above (risk-seeking) the true risk-neutral indifference curve as $\beta$ increases. The second effect is that risk-averse policies ($\beta = -1$ and $-0.5$) contain a large region of a stochastic choice profile that appears to depend only on the risky arm's parameter. We do not have a clear explanation for this effect. Our hypothesis is that risk-averse policies assume adversarial environments, which require playing mixed strategies with precise probabilities. Finally, the risk-neutral agent ($\beta = 0$) appears to be slightly risk-averse. We believe that this effect arises due to the noisy exploration policy employed during training.

## 5    Discussion

**Summary of contributions.**    In this work we have introduced a learning rule for the online estimation of the Gaussian free energy with unknown mean and precision/variance. The learning rule (2) is obtained by reinterpreting the stimulus-presence indicator component of the Rescorla-Wagner rule (Rescorla, 1972) as a (soft) indicator function for the event of either over- or underestimating the target value. In Lemma 1 we have shown that the free energy is the unique and stable fixed point of the expected learning dynamics. This is the main contribution.

Furthermore, we have shown how to use the learning rule for risk-sensitive RL. Since the free energy implements certainty-equivalents that range from risk-averse to risk-seeking, we were able to formulate a risk-sensitive, model-free update in the spirit of TD(0) (Sutton and Barto, 1990), thereby addressing a longstanding problem (Mihatsch and Neuneier, 2002) for the special case of the Gaussian distribution. Due to its simplicity, the rule is easy to incorporate into existing deep RL algorithms, for instance by modifying the error using a stop-gradient as shown in (23). In Section 3 we also elaborated on the role of the free energy within decision-making, pointing out its robustness properties and adversarial interpretation.

We also demonstrated the learning rule in experiments. Firstly, we empirically confirmed that the online estimates stabilize around the correct Gaussian free energies (Section 4–Estimation). Secondly, we showed how incorporating risk-attitudes into deep RL can lead to agents implementing qualitatively different policies which intuitively make sense (Section 4–RL). Lastly, we inspected the premia risk-sensitive agents are willing to pay for choosing a risky over a certain option, finding that agents have choice patterns that are more complex than we had anticipated (Section 4–Bandits).

**Limitations.** As shown empirically in Section 4–Estimation, an important limitation of the learning rule is that its fixed point is only equal to the free energy when the samples are Gaussian (or approximately Gaussian, as justified by the CLT). Nevertheless, agents using the risk-sensitive TD(0) update (11) still display risk attitudes monotonic in $\beta$, with $\beta = 0$ reducing to the familiar risk-neutral case.

While Lemma 1 establishes the stable equilibrium of the expected update, it only guarantees convergence in continuous-time updates. To show convergence using discrete-time point samples, a stronger result is required. In particular, we conjecture that

$$\left| J(v) \right| = 2 \left| \int N(x; \mu, \rho) \sigma_\beta (x - v)(x - v) \, dx \right| \leq 2 \left| \mathbf{F}_\beta - v \right| \tag{26}$$

If (26) is true, meaning that $J(v)$ is 2-Lipschitz, then this could be combined with a result in stochastic approximation theory akin to Theorem 1 in Jaakkola et al. (1994) to prove convergence.

A shortcoming of our experiments using R2D2 agents is that they deterministically pick actions that maximize the Q-value. However, risk-averse agents see their environments as being adversarial, and these in turn require stochastic policies in order to achieve optimal performance.

**Conclusions.** Because it is impossible to anticipate the many ways in which a dynamically-changing environment will violate prior assumptions, requiring the robustness of ML algorithms is of vital importance for their deployment in real-world applications. Unforeseen events can render their decisions unreliable—and in some cases even unsafe.

Our work makes a small but nonetheless significant contribution to risk-sensitivity in ML. In essence, it suggests a minor modification to existing algorithms, biasing valuation estimates in a risk-sensitive manner. In particular, we expect the risk-sensitive TD(0)-learning rule to become an integral part of future deep RL algorithms.

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
