# OpenReview forum: "Stochastic Approximation of Gaussian Free Energy for Risk-Sensitive Reinforcement Learning"
_NeurIPS.cc/2021/Conference — NeurIPS 2021 Submitted_

### Official Review · Reviewer_AeYb · 2021-07-16

**Rating:** 6
**Confidence:** 3

**Summary:**

This paper proposed an algorithm to estimate free energy. It provided a lemma to justify its update rule and conducted experiments to show the validity of its algorithm.

**Limitations And Societal Impact:**

The value of this algorithm (Why should we use it when we have other mature risk-sensitive RL algorithms? ) should be emphasized more.

**Main Review:**

Originality. The algorithm it proposed is an improvement of former algorithm. The intuition behind this improvement require further explanation. The lemma it proved is a basic property.

Quality. This work is complete and its result is supported. However, there might be some little mistake in it. For example, equation (5) should be double checked.

Clarity. This paper is written clearly.

Significance. I am not persuaded that it is significant by this paper. First, the author proposed using it for risk-sensitive RL. However, there are already several risk-sensitive RL algorithms. Why should we use free energy instead of other algorithm? Why is it superior? Secondly, the lemma here required Gaussian assumption and its only conclusion is that free energy is the fixed point. It did not include the rate of convergence.

**Time Spent Reviewing:**

5

---

> ### Author Response · Authors · 2021-08-10
> **Response to Reviewer 4**
>
> We thank the reviewer for their comments and suggestions.
>
> Issues:
>  1. (Originality): The Rescorla-Wagner rule is a learning rule for associative learning (which plays an important role in model-free RL, and has been used to e.g. model Pavlovian conditioning). In our work we deal with the problem of learning to act in a risk-sensitive manner. The result turns out to be indeed a clever modification to the Rescorla-Wagner rule - this is an interesting and non-trivial connection which was not obvious from the problem formulations alone. For the significance of using the free energy to model risk-sensitive decision-making see main response.
>  2. (Small mistake in (5)): We cannot spot a mistake in (5). Perhaps the reviewer can clarify?
>  3. (Why the free energy for risk-sensitive RL): See main response.
>  4. (No rate of convergence): See main response.

---

> > ### Comment · Reviewer_AeYb · 2021-08-14
> > **Thanks for your response**
> >
> > I thank the authors for their detailed response. I found that I misunderstood (5) and I apologize for my mistake.
> >
> > Concerning the free-energy formulation, you said that it is an old formal and unsolved treatment. However, what is its pros and cons when comparing with other risk-sensitive RL algorithms in practical might also be an interesting problem.
> >
> > Thanks again for your response! I will raise my score accordingly.

---

### Official Review · Reviewer_L68y · 2021-07-17

**Rating:** 4
**Confidence:** 5

**Summary:**

The authors introduced a stochastic approximation rule for estimating the free energy from
i.i.d. samples generated by a Gaussian distribution of unknown mean and variance. They apply their result then to optimize the entropic risk measure in an RL task.

**Limitations And Societal Impact:**

No potential negative impact is present.

**Main Review:**

## Originality:
At the best of my knowledge, the proposed update rule is indeed novel. The authors correctly give an overview of the state of the art related to the free energy, even if a related work section is missing.

## Quality:
The main result of the authors is Lemma 1, which seems to be correct. However it is not possible to apply it to  MDPs with a transition kernel which does not follow a Gaussian distribution. This limitation is recognized by the authors in a specific section, nevertheless, they show its use as a heuristic for TD learning.
They set a RL experiment on a gridworld task, in which the desired behaviour is obtained.
However, in order to prove that the heuristic is valid, it would have been nice to see it applied in a set of environments, with different difficulty levels. It would be interesting to have more general results about the update rule, for general distributions, or, at least, for some other families.

## Clarity:
The paper is clearly written, however, the structure is unusual: there is not a clear separation between introduction, related works, problem formulation and proposed methods. Organizing better the contents would help the reader figuring out the real contribution of the paper.

## Significance:
The result is interesting indeed, however, it is difficult to see if it will have an impact alone. There are indeed two problems that limits the application of the rule to RL:
- the Gaussian distribution assumption;
- the continuous time assumption.
The latter is necessary to prove the convergence to the fix point, however, RL is usually focused on discrete time settings.

**Time Spent Reviewing:**

2

---

> ### Author Response · Authors · 2021-08-10
> **Response to Reviewer 3**
>
> We thank the reviewer for their comments and suggestions.
>
> Issues:
>  1. (Larger set of environments, results for more general distributions): we chose our experiments to cover a set of stereotypical problems across different ML domains to make it easier for practitioners to relate to the theoretical results. We would be keen to hear concrete suggestions for additional tasks/environments that the reviewer envisions. Regarding theoretical results for more general distributions: we wholeheartedly agree that such results would be very interesting, however, at the current moment we do not see an obvious way to extend the theory to include non-Gaussian distributions (see our Limitations section where we sketch one such path, but currently lack the proof of a central conjecture).
>  2. (Structure of the paper is unusual): We chose to clearly state the main result first in order to contextualize the background. We will restructure our camera-ready version to make the contributions clearer.
>  3. (Gaussian assumption): We agree with the reviewer that this is a strong assumption (and we clearly point this out in our limitations section). Again, the parameter beta controls the equilibrium point (quantile of the distribution). Even if it does not match the free-energy in the non-Gaussian case, we can apply the method to general non-Gaussian domains to converge to quantiles of the distribution. See main response for more details.
>  4. (Continuous- vs. Discrete-time, convergence). We agree that a discrete-time proof of the convergence itself (the equilibrium point and its stability still being valid in the discrete case) would be best. See main response for more details.

---

> > ### Comment · Reviewer_L68y · 2021-08-19
> > **Reply**
> >
> > Thank you for answering my comments. About the environments: adding some proper noise to state-of-the-art benchmarks is tipically a good idea to obtain stochastic environments which are suitable to risk-averse optimization. For the moment I still think that the paper should be further improved in order to have either:
> > - a further theoretical grounding on the MDP side (see e.g. the suggestions by reviewer q5jG)
> > - or stronger empirical results that show experimentally the validity of the approach.

---

### Official Review · Reviewer_q5jG · 2021-07-20

**Rating:** 7
**Confidence:** 5

**Summary:**

This paper is about computing the log of the expectation an exp-transformed random variable (free energy) by stochastic approximation, under the assumption that this random variable is drawn from a Gaussian distribution. While not stated explicitely in this way, it seems that the underlying motivation may come from computation neuroscience: that of obtaining an explainable learning rule, in line the Rescorla-Wagner rule. I'm saying this because under the Gaussian assumption, the free energy admits a closed-form solution where the unknows are the mean and standard deviation which we don't necessarily need to estimate by stochastic approximation unless with have specic reasons to do so. The asymptotics behavior of the learning rule is studied in lemma 2 via an analysis akin to the "ODE method" in stochastic approximation: by considering the continuous time limit of the expected deterministic counterpart. Rather than talking about stability in terms of the eigenvalues of the Jacobian, the proof considers the possible values taken by the deterministic learning rule and shows that it has the free energy as a root. An application in reinforcement learning is presented where the authors match the form of their stochastic approximation method to that of temporal difference learning for policy evaluation and later present results in the control setting with deep reinforcement learning.

**Limitations And Societal Impact:**

I appreciate how figure 2 also shows how the algorithm performs when the Gaussian assumption is not satisfied.

See above comments on issues regarding the TD case

**Main Review:**

This paper is impeccably writen: the flow, clear notation and figures (latex-styled plots, pdf output), historical context and cited papers older than a year ago. This really sets the standard for what our community should be striving to achieve. It is really refreshing to read a paper structured like this one! It starts right from the begining with a clear and direct description of what has been found, in math. The high level motivation then follows. It's not wasting anyone's time trying to sell their work. I wouldn't be surprise to learn that the first author is a physicist who moved to computational neuroscience.

# Issues

- The experiments sections departs quite far from the original setup where the assumptions are clear and the analysis is tractable. It feels like those experiments (the deep rl ones) were designed to make the paper feel more "neurips-y" and appeal to a more ML-focused audience. I would have preferred to see simpler but more carefully designed experiments. More on that in the comments below

- Regarding the application to TD, of course many questions are left unanswered. Note that you start with the policy evaluation setup and then go to the control setting in the experiments. Let's just start with policy evaluation: in the tabular setting, is the corresponding method stable? What could you say via the ODE method? Can you show that the corresponding deterministic operator is a contraction? If not, can you establish stability in a different way? Now one step further: in the policy evaluation setting, what happens if now throw in linear functiona approximation? Do you obtain a system of "projected Bellman equations"? And finally the control setting: take Q learning in the tabular setting, add your update rule, does it converge? What do you get when you apply the ODE method once more (see Kushner and Yin for Q learning)?

(Reflecting on the above, perhaps nothing could be said due to the iid assumption.)

# Notes

Line 37: "is defined" it'd be nice to have a citation

Equation 7: Time indices would be appropriate here and throughout the entire paper: $v_{t+1} = v_t + \alpha_t \cdot (x_t - v_t)$.

Line 153: It doesn't look like the analytical error function (21) would lend itself easily to automatic differentiation due to the presence of the dilogarithm term. Tensorflow seems to offer tf.math.special.spence, but what is it really going on under the hood? What is the corresponding vector-Jacobian rule?

Line 183: "trained R2D2" The experimental setup and the choice of figures is appropriate. However, using R2D2 seems to be overkill for the problem being solved here. I would have preferred to see a minimalistic vanilla Q-learning + SA Free energy demonstration than a deep RL setup for a grid world problem.

Line 208: "memory-based meta-learning" again, the experimental setup seems overkill and clashes with the more methodical approach taken earlier in the paper.

Line 239: "This is the main contribution." I love this

**Time Spent Reviewing:**

3

---

> ### Author Response · Authors · 2021-08-10
> **Response to Reviewer 2**
>
> We thank the reviewer for their comments and suggestions.
>
> Issues:
>  1. (Experimental section departs from orig. setup). Our first experiment is in line with the theory, and also shows how the method behaves under uniform samples: it still chooses quantiles that are consistent with risk-sensitive decision-making, although they are numerically different from the free energy. Our other two experiments were selected to reinforce this point, highlighting that using the method as a heuristic still leads to risk-sensitive behavior. We chose the water-corridor and bandit tasks because we thought they would appeal to an RL and a decision-making audience.
>  2. (ODE analysis). These are great suggestions which clearly haven’t been addressed in the paper. We understand the suggested steps, using the ODE to derive convergence and rate of convergence, but right now we admittedly only have a vague intuition about what to expect using this method and we thought it could be better left for a follow-up paper.
>
> Comments:
>  * Line 37: “Probability Models for Economic Decisions”, 2nd Edition, by Myerson and Zambrano (2019).
>  * Line 153 (what is going on under the hood, tensorflow auto-diff seems to run into issues with the dilogarithm term): Good observations. The error term was derived to explain the structural connection to the quadratic error, but of course we suggest using a stop-gradient as in (23) for practical implementations.
>  * Line 183 (use of R2D2): We also have Q-learning simulations which we are happy to provide in a camera-ready version.
>  * Line 208 (meta-learning for bandits is overkill): We agree meta-learning appears to be overkill. However, we wanted to derive the Bayes-optimal policy, i.e. the one that explores-exploits optimally by definition (and not in the “big-O regret” sense, as in, say, UCB or Thompson Sampling). Meta-learning was actually the simplest and most robust way of approximating the problem numerically we could think of.
>  * Line 239: Thanks!

---

### Official Review · Reviewer_sDnj · 2021-07-20

**Rating:** 7
**Confidence:** 4

**Summary:**

The authors provide a novel approximation rule for the risk-sensitive criterion in RL. In particular they present an iterative update rule for the estimation of the free energy, which has direct application to risk-sensitive RL. They point out relevant properties of the approach and their relationship to game-theoretic and finance and evaluate these on a set of benchmark problems.

**Limitations And Societal Impact:**

yes

**Main Review:**

The paper is written very well.  While heavy on theory it is explained in an understandable way and the resulting methodology is simple.

The experiments make sense given the research question. I particular like that the authors start with simple toy problems (including verifying the estimator), before using to more complex domains such as 2-armed bandits


Cons:
As the authors point out the main limitation is the Gaussian assumption for the Returns.  In this case the paper would have been stronger had the authors constructed an (toy) experiment where the Gaussian assumption does not hold, ideally with long-tail reward distributions, as these are often seen in practice.

Disclaimer: I did not check the math in Section 2, so for the review I will assume it to be correct.

A question: The free energy as certainty-equivalent applied to RL always appeared a bit odd to me: If we have the expected return (units return) we are adding the variance (units squared return) to it, weighted by  \beta.  This does not seem right. Wouldn't it make more sense to use \mu + \beta \sigma here (as the standard deviation and the mean share the same unit)?  Could from your update rule also an analogous rule for this be derived?

**Time Spent Reviewing:**

3

---

> ### Author Response · Authors · 2021-08-10
> **Response to Reviewer 1**
>
> We thank the reviewer for their comments and suggestions.
>
> Issues:
>  1. Heavy-tailed toy experiment: Thank you for the great suggestion. We will provide such an example for the camera-ready version, using a Student’s t-distribution with one degree of freedom.
>  2. Why the variance (which has squared-reward units), and not the std-dev (which has reward units)? We understand your intuition: if we take $\mu + \beta \sigma$, then $\beta$ is a dimensionless quantity that picks out a scale-invariant quantile/confidence interval; whereas in $F = \mu + \frac{1}{2} \beta \sigma^2$, $\beta$ has the same dimension as $\frac{1}{\sigma}$, so the choice of $\beta$ is linked to the dimensions of the coordinate system, which seems strange at first. However, in the free energy, $\beta$ plays a different role: it is the conversion factor from “units of information” to “value”, so $\beta$ has units of “value/information”. In other words, $\beta$ is converting the bits spent (on the risk-sensitive transformation of the reference Gaussian, which has $\sigma^2$ units) into units of value.

---

### Author Response · Authors · 2021-08-10
**Response to all reviewers**

We thank the reviewers for thoroughly reviewing our manuscript and the helpful comments. We are happy to hear that all reviewers agree that
 * the main idea is novel and original
 * the derivations and results are correct,
 * and the manuscript is well written and the results are interesting.
All reviewers agree with the main limitations of the work, which we clearly pointed out and discussed in the manuscript.

We want to reiterate that the free energy (=“exponential rewards”) formalization was actually one of the first formal treatments of risk-sensitive decision-making, going at least as far as Howard and Matheson (1972) fifty years ago. How to incorporate this into a model-free update has been an unsolved problem since then (see Mihatsch and Neuneier 2002), and here we provide a solution for the gaussian case.

Reviewers 3 and 4, who were most critical, point out a potential lack of significance due to the Gaussian assumption required for our theoretical results to hold. We agree that estimating the free energy for a broader class of distributions would be desirable, but we believe this cannot be done with a model-free approach. A significantly harder distributional- or model-based approach would work.

It is worth noting that our risk-sensitive TD-learning rule is generally applicable to non-Gaussian cases (in spite of not being an exact approximation to the free energy), and discrete-time cases. We believe that its extreme simplicity will be of significant interest to RL practitioners, particularly because it captures both risk-seeking and risk-averse behavior using a small modification of the standard rule. We are not aware of many approaches that handle both cases seamlessly.

Concerning convergence, the continuous time assumption makes it possible to provide a simple proof by only considering the equilibrium point and its stability. For the discrete-time case, we now have a proof (which we will add to the camera-ready) that the update function J is Lipschitz-continuous with constant ~1.15. It is known via the result from Dvoretzky, 1956 (section 8) that having a constant < 1 exactly would prove convergence. We are working on extending the proof to our case, but have not managed to finish this  within the rebuttal period. Moreover, we conjecture that L2 convergence would be possible if the learning rate follows the correct schedule (diverging series but bounded squared series), and that in our experiments, with a constant learning-rate, we would get L1 convergence (variance not converging to 0 at infinity).

More detailed comments are addressed in individual reviewers’ responses.

---

### Decision · Program_Chairs · 2021-09-27

**Decision:**

Reject

**Comment:**

The paper introduces some technical ideas that may interest a significant segment of the community.  The theoretical work is carried out under quite restrictive assumptions.  But the ideas may be of broader interest.